REVIEW-SYMPOSIUM

# The role of oestrogen in determining sexual dimorphism in energy balance

Anne Nicole De Jesus and Belinda A. Henry 

*Metabolism, Obesity and Diabetes Program, Biomedicine, Discovery Institute, Department of Physiology, Monash University, Clayton, Victoria, Australia*

Handling Editors: Laura Bennet & Janna Morrison

The peer review history is available in the Supporting information section of this article (https://doi.org/10.1113/JP279501#support-information-section).

**Abstract** Energy balance is determined by caloric intake and the rate at which energy is expended, with the latter comprising resting energy expenditure, physical activity and adaptive thermogenesis. The regulation of both energy intake and expenditure exhibits clear

**Anne Nicole De Jesus** is a doctoral student under the supervision of Dr Belinda Henry. She received a double degree BA extended major in Psychology (2019) and BSc (Hons) major in Physiology (2020) at Monash University. Her current work explores the role of female sex hormones on brown fat activity and glucose metabolism in women across different stages of reproductive live. **Belinda Henry** is the Director of the Biomedicine Discovery Institute Graduate School and a Senior Lecturer within the Department of Physiology at Monash University, Australia. She is the head of the Metabolic Neuroendocrinology Group. Her research centres on the discovery of novel means of altering energy expenditure through modulation of the brain as a mechanism for prevention or treatment of obesity. Her work is particularly focused on the process of thermogenesis and the mechanisms that underpin this process in brown adipose tissue and

skeletal muscle. She has strong expertise in utilising large animal models for neuroendocrine research and she has recently established the Translational Metabolic and Exercise laboratory.

This review was presented at the Australian Physiological Society Diamond Jubilee Conference 'Sex differences in Physiology', which took place at Griffith University, Gold Coast QLD, Australia, 22 November 2021.

sexual dimorphism, with young women being relatively protected against weight gain and the development of cardiometabolic diseases. Preclinical studies have indicated that females are more sensitive to the satiety effects of leptin and insulin compared to males. Furthermore, females have greater thermogenic activity than males, whereas resting energy expenditure is generally higher in males than females. In addition to this, in post-menopausal women, the decline in sex steroid concentration, particularly in oestrogen, is associated with a shift in the distribution of adipose tissue and overall increased propensity to gain weight. Oestrogens are known to regulate energy balance and weight homeostasis via effects on both food intake and energy expenditure. Indeed, $17\beta$-oestradiol treatment increases melanocortin signalling in the hypothalamus to cause satiety. Furthermore, oestrogenic action at the ventromedial hypothalamus has been linked with increased energy expenditure in female mice. We propose that oestrogen action on energy balance is multi-faceted and is fundamental to determining sexual dimorphism in weight control. Furthermore, evidence suggests that the decline in oestrogen levels leads to increased risk of weight gain and development of cardiometabolic disease in women across the menopausal transition.

(Received 2 May 2022; accepted after revision 26 July 2022; first published online 18 September 2022)

**Corresponding author** B. A. Henry: Metabolism, Obesity and Diabetes Program, Biomedicine Discovery Institute, Department of Physiology, 26 Innovation Way, Monash University, Clayton, VIC 3800, Australia. Email: belinda.henry@monash.edu

**Abstract figure legend** A schematic illustration highlighting the key role of oestrogen in the protection against weight gain and cardiometabolic disease in young reproductive age women. High concentrations of oestrogens in young women are associated with differences in adipose tissue distribution, such that adipose is primarily accumulated in the gynoid subcutaneous regions compared to the android adipose phenotype typically seen in men and post-menopausal women. In addition, circulating levels of oestrogen in women of reproductive age are associated with inherent changes in energy expenditure including increased physical activity and adaptive thermogenesis. These differences in body composition and energy balance are thought to underpin relative resistance to weight gain and improved metabolic health in young women.

## Introduction

Excess adiposity and obesity increase the risk of developing cardiometabolic diseases, which include hypertension, stroke, insulin resistance and type 2 diabetes (T2D) (Khan et al., 2018; Mokdad et al., 2003). Thus, overweight and obesity are associated with significantly increased risk of morbidity and mortality. In the simplest sense, obesity is caused by the imbalance in energy homeostasis whereby food intake exceeds energy expenditure; however, this does not account for the complex and multifactorial mechanisms that are essential to the control of body weight (Di Tecco et al., 2020; Salinero et al., 2018). Previous work has shown clear sexual dimorphism in the prevalence of obesity as well as the development of hypertension, glucose intolerance and type II diabetes. Women have evolved to typically have overall increased adiposity compared to men (Schorr et al., 2018). Despite this, pre-menopausal women are considered to be relatively protected against weight gain and the development of cardiometabolic diseases compared to age-matched men (Schorr et al., 2018). This protection against cardiovascular and metabolic disease is at least partly due to differences in body composition and the distribution of adipose tissue, where young women typically display adipose accumulation within the subcutaneous lower body regions (gynoid body shape) (Fried et al., 2015; Karastergiou et al., 2012). On the other hand, men display an android distribution of adipose tissue, which is characterised by accumulation within the visceral compartments (Fried et al., 2015; Karastergiou et al., 2012). This disparity in the distribution of adipose tissue is primarily driven by differing concentrations of sex steroids. Indeed, gender-affirming hormone therapy can modulate the android: gynoid adipose ratio, where oestrogen therapy lowers the android: gynoid adipose ratio in transgender women and androgen therapy increases the android: gynoid ratio in transgender men (Bretherton et al., 2021). In addition, across the menopausal transition and with declining levels of ovarian steroids, women experience a shift in the distribution of adipose tissue from the gynoid to the android phenotype (Greendale et al., 2021) and this is clearly associated with increased risk of developing cardiometabolic diseases (Sari et al., 2019). It is important to highlight that not only is menopause associated with a change in body conformation, but women also experience

generalised weight gain (Davis et al., 2012; Proietto, 2017), and it has been reported that overweight/obese women gain more weight in midlife than their lean counterparts (Proietto, 2017). However, to date, the mechanisms that underpin weight gain in women across mid-life remain largely unknown.

Body weight and adiposity are primarily determined by caloric intake and the rate at which energy is expended. Energy expenditure has three major components: resting energy expenditure (REE)/basal metabolic rate (BMR), physical activity and adaptive thermogenesis. Numerous studies have shown sex differences in the regulation of both food intake and energy expenditure, with effects being both dependent and independent of sex steroid action. A good example of sexual dimorphism is that females are more responsive to the adipose-derived hormone leptin and have enhanced melanocortin signalling in the brain, which may confer increased satiety (Clegg et al., 2006; Quarta et al., 2021; van Veen et al., 2020; Wang et al., 2018). To date, studies have also suggested that sex differences may result from inherent differences in developmental programming, X-linked chromosomal genes, neuroanatomical differences of hypothalamic feeding circuitry as well as variations in sex steroids (Wang & Xu, 2019). In regard to the latter, there is strong evidence to suggest that oestrogens play a vital role in regulating body weight and metabolic function. There are three main oestrogenic hormones, oestriol, $17\beta$-oestradiol and oestrone, with $17\beta$-oestradiol being the predominant steroid synthesised in pre-menopausal women (Cui et al., 2013). Indeed, studies have demonstrated that the beneficial metabolic effects of oestrogens are primarily via $17\beta$-oestradiol action. In particular, $17\beta$-oestradiol is known to exert reciprocal effects to reduce food intake and increase energy expenditure. This review will discuss the role of oestrogen in determining sexual dimorphism in the control of energy balance, with a particular emphasis on the regulation of energy expenditure.

## Sexual dimorphism and the role of oestrogen in determining weight gain: evidence from pre-clinical models

Evidence shows that one's susceptibility to weight gain is determined by myriad genetic and physiological factors. Adult male mice have a greater propensity to gain weight, irrespective of diet, compared to their female counterparts (Hong et al., 2009). This protective effect against weight gain in females has primarily been attributed to circulating levels of oestrogen, since ovariectomy abolishes the sex differences seen in weight gain, with adult female mice gaining weight to a similar degree to males (Hong et al., 2009). Indeed, oestradiol benzoate treatment to ovariectomised female mice has

been shown to prevent ovariectomy-induced weight gain (Cavalcanti-de-Albuquerque et al., 2014). A further example of the importance of oestrogen in the control of body weight is the aromatase knockout (ArKO) murine model, whereby deletion of the enzyme responsible for converting androgens to oestrogens leads to obesity in male and female mice (Jones et al., 2000). It is also important to note that in general male mice are not only more susceptible to weight gain than female mice, but also exhibit an altered inflammatory profile as well as impaired insulin sensitivity, which increases the risk of developing secondary cardiometabolic complications including hypertension and T2D (Estrany et al., 2013). Thus, evidence suggests that murine models are similar to humans (as described above in Introduction) whereby circulating oestrogens exert numerous protective effects on metabolic health.

Oestrogenic actions are exerted via the nuclear oestrogen receptors $\alpha$ and $\beta$ (ER$\alpha$ and ER$\beta$) as well as the cell membrane G-protein-coupled oestrogen receptor (GPER). With regard to the classical nuclear receptors, studies in rodents indicate that the beneficial effects of oestrogen on body weight and metabolic function are primarily mediated via ER$\alpha$ (Roesch, 2006). Genetic deletion of ER$\alpha$ in both male and female mice leads to weight gain, insulin resistance and glucose intolerance (Heine et al., 2000). Furthermore, the central nervous system is integral to the protective metabolic effects of oestrogens, since brain-specific deletion of ER$\alpha$ leads to obesity and increased adiposity in female mice, despite a compensatory increase in peripheral concentrations of $17\beta$-oestradiol (Xu et al., 2011). It is now widely recognised that $17\beta$-oestradiol acts at various sites within the central nervous system to reduce food intake and increase energy expenditure, which will be discussed in detail below.

In addition to the classical ER$\alpha$, metabolic effects of oestrogen can also occur via GPER, previously referred to as GPR30 (Saito et al., 2015; Sharma & Prossnitz, 2021). In rodents, GPER is abundantly expressed in metabolically important tissues including adipose tissues, pancreatic islets and the hypothalamus (Davis et al., 2014; Hazell et al., 2009; Prossnitz et al., 2007; Shaklai et al., 2019). Indeed, a recent study demonstrated immunostaining of GPER within neurons, astrocytes and oligodendrocytes in the arcuate nucleus, paraventricular nucleus, lateral hypothalamus and ventromedial nucleus of the hypothalamus in male and female rats (Marraudino et al., 2021). Moreover, levels of immunostaining revealed sexual dimorphism whereby the total number of GPER-immunostained cells was greater in the arcuate nucleus and the lateral hypothalamus of female than male rats (Marraudino et al., 2021). Despite this, genetic knockout studies have produced dichotomous findings with some studies showing that deletion of the GPER in

mice can produce a similar metabolic phenotype to the ERα knockout mice with the onset of obesity and insulin resistance (Sharma & Prossnitz, 2021). Alternatively, studies have demonstrated that female GPER knockout mice have lowered body weights compared to males, which contrasts with the known effects of oestrogen to reduce weight gain (Wang et al., 2016). Thus, further work is required to delineate the physiological role of GPER in relaying the metabolic effects of 17β-oestradiol.

### Neuroendocrine control of food intake

Various endocrine factors including leptin, ghrelin, insulin and 17β-oestradiol are integral to the control of food intake and energy expenditure (Briggs & Andrews, 2011; Cowley et al., 2001; Dodd et al., 2015; Gao et al., 2007; Wallen et al., 2001). The neural control of food intake involves the complex interplay between both reward and homeostatic pathways (Simon et al., 2017). The hypothalamus is known to be essential to the integration of information provided by peripheral signals to effect physiological control of energy balance. The satiety hormones, leptin and insulin, act at the proopiomelanocortin (POMC) and neuropeptide Y/agouti-related protein (NPY/AgRP) neurons in the arcuate nucleus (Fig. 1). Both leptin and insulin depolarise POMC neurons leading to the release of the neurotransmitter α-melanocyte-stimulating hormone (α-MSH), which reduces food intake via the melanocortin-4 receptor (MC4R)-expressing neurons located in the paraventricular nucleus (Cowley et al., 2001) (Fig. 1). In contrast, NPY/AgRP neurons in the arcuate nucleus are responsible for signalling hunger (Willesen et al., 1999) (Fig. 1). AgRP acts as an endogenous antagonist to inhibit α-MSH binding at the MC4R and thus prevents the satiety signal from the POMC neurons (Pritchard et al., 2004) (Fig. 1). NPY increases food intake via the Y1 and Y5 receptors located on neurons within the paraventricular nucleus (Dube et al., 2000; Duhault et al., 2000; Pralong et al., 2002) (Fig. 1). Leptin and insulin directly activate POMC neurons and inhibit NPY/AgRP neurons, thus reducing food intake and causing satiety (Fig. 1). On the other hand, the hunger hormone, ghrelin, is secreted from X/A-like cells of the stomach immediately prior to a meal (Stengel & Tache, 2012) and directly activates NPY/AgRP neurons (Fig. 1). It is important to note that factors that regulate food intake exert reciprocal effects to also regulate energy expenditure. For example, leptin acts at the brain to cause satiety but also increases energy expenditure, thus effecting weight loss in lean individuals (Henry et al., 2008, 2011; Heymsfield et al., 1999). Interestingly, pre-clinical studies have demonstrated that 17β-oestradiol acts in concert with leptin to regulate food intake, energy expenditure and body weight.

**Sexual dimorphism in the control of food intake: the interplay between leptin and oestrogen.** There are clear sex differences in the neuroendocrine control of food intake. Various animal studies, in both rodents and sheep, have shown that the satiety effect of leptin is greater in

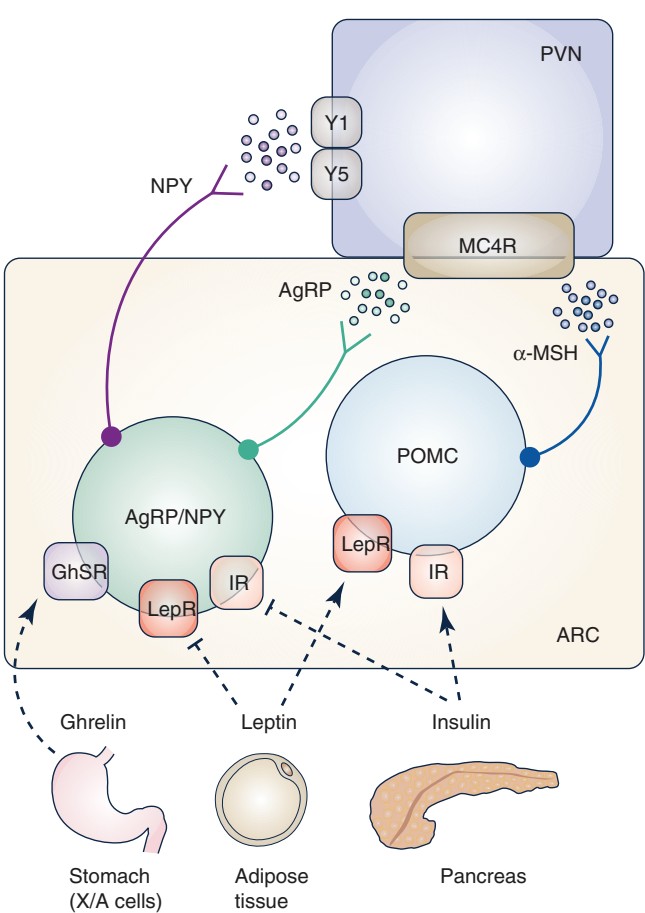

**Figure 1. The hypothalamic subnuclei involved in the regulation of the food intake pathway**
The ARC houses the AgRP/NPY and POMC neurons. Leptin is released from adipose tissues and acts on the leptin receptors of the POMC neurons. On the other hand, insulin is released from the pancreas and acts on the insulin receptors of the POMC neurons. The activation of LepR and IR results in the inhibition of the AgRP/NPY neurons and thus decreased feeding. Ghrelin, released from the X/A-like cells of the stomach, acts upon the GhSR located on the AgRP/NPY neurons, which results in the release of AgRP, acting as an endogenous antagonist against α-MSH. NPY is also released, which acts on Y1/Y5 receptors located within the PVN, and alongside the effects of AgRP on MC4R, promote feeding. α-MSH, α-melanocyte stimulating hormone; AgRP, agouti-related peptide; ARC, arcuate nucleus; E2, 17β-oestradiol; ERα, oestrogen receptor α; GhSR, ghrelin receptors; IR, insulin receptors; LepR, leptin receptors; MC4R, melanocortin-4 receptor; NPY, neuropeptide Y; POMC, pro-opiomelanocortin; PVN, paraventricular nucleus; Y1, Y1 receptor; Y5, Y5 receptor.

females than males (Clarke et al., 2001; Clegg et al., 2006; Rosenbaum et al., 2001; Saad et al., 1997). Indeed, in rats, 17$\beta$-oestradiol treatment to gonadectomised males increases leptin sensitivity so that the leptin-induced decrease in food intake is similar to that seen in intact females (Clegg et al., 2006). Similarly, the effect of insulin and ghrelin on food intake is sexually dimorphic, with the satiety effect of insulin being greater in female than male rats, whereas the orexigenic effect of ghrelin is exacerbated in male compared to female rats (Clegg et al., 2006); importantly these effects have been attributed to differences in the circulating levels of 17$\beta$-oestradiol. Thus, it has been suggested that a degree of cross-talk exists between the mechanisms that underpin the satiety effects of oestrogens and classical metabolic hormones such as leptin and ghrelin. Despite this, a recent study demonstrated that oestradiol benzoate treatment of ovariectomised female mice had no effect on the ability of leptin to increase signal transducer and activator of transcription 3 (STAT3) phosphorylation (Kim et al., 2016). This indicates that oestrogen does not regulate body weight simply via enhancing leptin action. Instead, studies strongly suggest that 17$\beta$-oestradiol directly regulates body weight via ER$\alpha$-expressing neurons within the hypothalamus. Oestrogen exerts numerous actions via ER$\alpha$-expressing neurons in the hypothalamus, including but not exclusive to, control of reproductive function and the hypothalamo–pituitary–gonadal axis, sexual behaviours, appetite and energy expenditure (Gieske et al., 2008; Martinez de Morentin et al., 2014; Santollo et al., 2007; Trouillet et al., 2022; van Veen et al., 2020). The latter two components will be discussed in detail below.

In both animals and humans, endogenous fluctuations in circulating 17$\beta$-oestradiol levels have been linked to cyclical changes in food intake in females (Lyons et al., 1989). In women, food intake is lowest during the peri-ovulatory phase of the menstrual cycle, when 17$\beta$-oestradiol levels are highest (Kemnitz et al., 1989; Lyons et al., 1989). Oestrogenic effects to reduce food intake are primarily mediated via ER$\alpha$, as the satiety effect of 17$\beta$-oestradiol treatment is abolished in ER$\alpha$ knockout mice (Santollo et al., 2007). Further studies in mice suggest that the satiety effect of 17$\beta$-oestradiol is primarily via the hypothalamic melanocortin pathway (Ainslie et al., 2001; Geary, 2001; Hirosawa et al., 2008; Roepke et al., 2010; Stincic et al., 2018). In female mice, altered expression of *Agrp* and *Npy* in the arcuate nucleus coincides with fluctuations in food intake across the ovarian cycle and genetically induced neurodegeneration of AgRP neurons abolishes cyclical changes in feeding (Olofsson et al., 2009). However, this effect appears to be indirect as, at least in mice, ER$\alpha$ is not co-localised to the AgRP/NPY neuron (Olofsson et al., 2009). On the other hand, specific deletion of ER$\alpha$ in POMC neurons of the arcuate nucleus causes hyperphagia and obesity in female mice (Xu et al., 2011) and more recent studies suggest that specific subpopulations of POMC neurons may be integral to determining sex differences in energy balance (Hubbard et al., 2019; Quarta et al., 2021; Reilly et al., 2019; Wang et al., 2018).

Within the arcuate nucleus, POMC neurons are known to be heterogeneous, with studies suggesting that discrete topographical subpopulations may differentially regulate food intake, energy expenditure as well as peripheral glucose and lipid metabolism (Quarta et al., 2021). Female mice are known to have a greater number of POMC neurons than male mice (Wang et al., 2018) and genetic manipulation of POMC neuronal subpopulations leads to sexually dimorphic control of body weight in male and female mice (Wang et al., 2018). For example, deletion of the transcription factor Tap63 from POMC neurons elicits weight gain and obesity in female mice fed a high-fat diet and thus abolishes the innate protection offered by oestrogen; there is little effect on weight gain in male mice (Wang et al., 2018). Interestingly, two studies have recently suggested that POMC neurons exert sexually divergent effects to preferentially regulate energy expenditure in females *versus* food intake in males. In mice, targeted mutation of the *Pomc* gene to abolish the synthesis of both $\alpha$-MSH and desacetyl-$\alpha$-MSH causes hyperphagia in males, but in contrast, reduces energy expenditure in females (Hubbard et al., 2019). Similarly, the deletion of Gpr17 in POMC neurons causes significantly less weight gain in females due to enhanced locomotion and increased energy expenditure compared to males (Reilly et al., 2019). This highlights an important role for energy expenditure in determining the effects of sex steroids on energy homeostasis as well as the effects of sexual dimorphism in the control of body weight.

Although oestrogens are known to cause satiety, it is important to highlight that the effects of 17$\beta$-oestradiol on food intake are frequently considered to be transient. In ovariectomised ewes, chronic 3 × 3 cm silastic implants containing 17$\beta$-oestradiol causes an initial decrease in food intake; however, this returns to baseline within 5 days of treatment (Clarke et al., 2013). Furthermore, genetic deletion of the aromatase enzyme in male and female ArKO mice causes weight gain, which manifests in the absence of hyperphagia (Jones et al., 2000). To further highlight the possible role of energy expenditure in mediating oestrogen-induced changes in body weight, Rogers et al. (2009) demonstrated that ovariectomy-induced weight gain across 12 weeks was associated with reduced energy expenditure without an associated change in daily food intake (Rogers et al., 2009). This suggests that in the long-term, oestrogens and in particular 17$\beta$-oestradiol, acts to regulate body weight primarily via the modulation of energy expenditure.

## Mechanisms of energy expenditure

As indicated above, there are three major forms of energy expenditure, namely REE or BMR, physical activity, and adaptive thermogenesis. The latter refers to the dissipation of energy through specialised heat production and occurs in brown and beige adipocytes as well as skeletal muscle (Gong et al., 1997; Hilse et al., 2016; Walden et al., 2012). Thermogenesis in brown and beige adipocytes is regulated by both central and peripheral factors. Typically, thermogenic stimuli, including exposure to low ambient temperature or food consumption, are perceived by the brain leading to activation of the sympathetic nervous system. In turn, noradrenaline via action at $\beta$-adrenergic receptors on brown adipocytes causes activation of uncoupling protein (UCP)-1, which creates a proton leak across the inner mitochondrial membrane; this process of uncoupled respiration leads to the dissipation of energy via heat production (Fig. 2). Exogenous factors including peroxisome proliferator-activated receptor $\gamma$ (PPAR$\gamma$) agonists and $\beta_3$-adrenergic agonists can increase thermogenesis via activation of brown adipocytes or recruitment of beige adipocytes, which occurs through the process known as 'browning' (Petrovic et al., 2010). In addition to adipose (brown or beige) thermogenesis, myocytes are capable of producing heat through similar futile cellular pathways involving both mitochondrial uncoupling through UCP3 and futile calcium cycling (Bal & Periasamy, 2020; Clarke et al., 2012).

To date, various studies have indicated that the physiological control of energy expenditure is sexually dimorphic; however, the direction of this effect is dependent on the specific component of energy expenditure. For example, REE is greater in men than women, but on the other hand, women exhibit increased thermogenesis compared to men (Arciero et al., 1993; Au-Yong et al., 2009; Cypess et al., 2009; Ferraro et al., 1992; Fletcher et al., 2020; Kirkby et al., 2004; Ouellet et al., 2011). Furthermore, pre-clinical data from murine models clearly demonstrate that oestrogens act at the ventromedial hypothalamus (VMH) to regulate multiple facets of energy expenditure including both physical activity and adaptive thermogenesis (Krause et al., 2021; Martinez de Morentin et al., 2014; van Veen et al., 2020).

**Sexually dimorphic control and the impact of menstrual cyclicity on resting energy expenditure.** REE is defined as the energy expended to maintain physiological functions essential to life, and this contributes to approximately $60-70\%$ of total daily energy expenditure (Blundell et al., 2012). The primary determinant of REE is lean body mass, and as such, men typically have a higher rate of REE than women (Arciero et al., 1993; Rodriguez et al., 2002; Souba, 1997). In addition to this frank sexual dimorphism, a recent systematic review provided evidence to suggest that there is a small effect of menstrual cyclicity on REE in pre-menopausal women (Benton et al., 2020). In this case, REE may be slightly increased in women during the luteal phase compared with the follicular phase of the menstrual cycle (Benton et al., 2020). Whether this coincides with fluctuations in sex steroids remains unknown and requires further investigation. Additionally, a reduction in REE has been reported across the menopausal transition, and REE is thought to be lower in post-menopausal women due to a decline in lean mass (Van Pelt et al., 1997). Despite this, in young women, suppression of sex steroid levels by chronic administration of a gonadotropin-releasing hormone (GnRH) agonist (for 5 months) decreased REE; this effect was attenuated by 17$\beta$-oestradiol replacement (Melanson et al., 2015). This suggests that oestrogen may exert some effect to regulate energy expenditure via modulating metabolic rate, at least in young women of reproductive age.

**Physical activity is regulated via oestrogen in a sexually dimorphic manner.** Physical activity comprises both spontaneous activity or non-exercise activity thermogenesis and volitional exercise. Various endogenous and exogenous factors are known to influence levels of physical activity including genetic variance, biological determinants (including exercise ability and motivation), as well as social-environmental factors such as diet

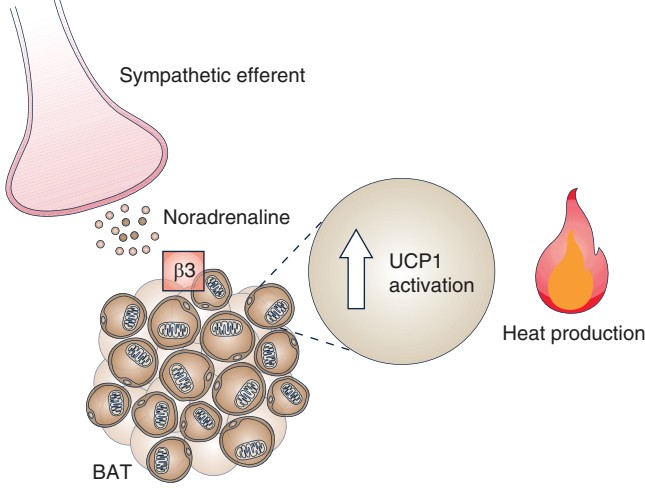

**Figure 2. The activation of adaptive thermogenesis within brown adipose tissues**
The activation of the sympathetic nervous system results in the release of noradrenaline, which is a metabolic regulator of brown adipose tissues. A series of events activate UCP1, which generates heat and contributes to energy expenditure via adaptive thermogenesis. ARC, arcuate nucleus; BAT, brown adipose tissue; $\beta$3, $\beta_3$-adrenoceptor; UCP1, uncoupling protein-1.

(Lightfoot et al., 2018). Indeed, sex steroids are considered to be a biological factor that can influence physical activity (Lightfoot et al., 2018). Studies conducted using rodent models show that ovariectomy causes a marked decrease in spontaneous physical activity, which can be reversed by 17$\beta$-oestradiol treatment (Gorzek et al., 2007; Izumo et al., 2012; Krause et al., 2021). In addition, longitudinal studies have demonstrated that post-menopausal women are less likely to engage in physical activity than the same individuals prior to the menopausal transition (Poehlman, 2002). To further elucidate the role of oestrogen in regulating physical activity in women, Melanson et al. (2015) chronically reduced ovarian steroid levels via administration of a GnRH agonist, which led to a reduction in exercise-associated energy expenditure; this effect was prevented by 17$\beta$-oestradiol treatment. Together these data indicate that oestrogen may exert effects, at least in women, to enhance physical activity. To further support this, work in mice indicates that oestradiol acts at the VMH to regulate energy expenditure, via increased physical activity and adaptive thermogenesis (Koch, 1990; Krause et al., 2021; van Veen et al., 2020) (Fig. 3). Chemogenic activation of ER$\alpha$-expressing neurons in the VMH leads to an increase in spontaneous activity in both male and female mice (van Veen et al.,

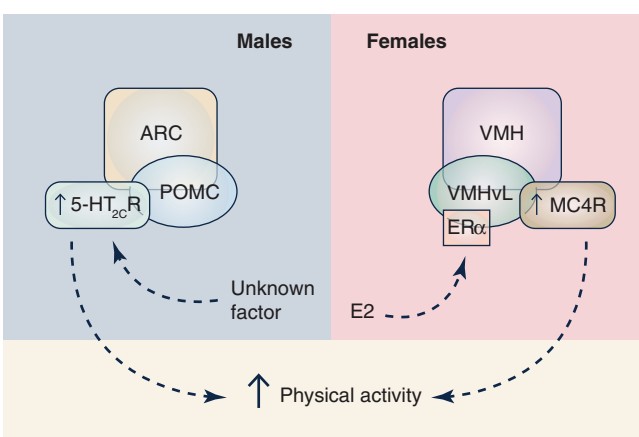

**Figure 3. The hypothalamic subnuclei involved in the regulation of physical activity in males and females**
It is proposed that a subpopulation within the VMH, known as the VMHvl, co-expresses both ER$\alpha$ and MC4R. 17$\beta$-Oestradiol activates the VMHvl, which leads to the upregulation of the MC4R, thereby promoting physical activity in females. In males, POMC within the ARC promotes physical activity via the activation of the 5-HT$_{2C}$R, which is a serotonin receptor, regulated by an unknown factor. 5-HT$_{2C}$R, 5-hydroxytryptamine 2c receptor; ARC, arcuate nucleus; E2, 17$\beta$-oestradiol; ER$\alpha$, oestrogen receptor $\alpha$; MC4R, melanocortin-4 receptor; POMC, pro-opiomelanocortin; VMH, ventromedial hypothalamus; VMHvl, ventrolateral ventromedial hypothalamus.

2020). However, the VMH is known to be a sexually dimorphic nucleus, and consistent with this, there are discrete subpopulations of neurons that are differentially involved in the regulation of physical activity in males and females (Krause et al., 2021; van Veen et al., 2020) (Fig. 3). In female mice, ER$\alpha$ is abundantly expressed in the ventrolateral VMH, with lower expression in males (Koch, 1990). Furthermore, oestradiol benzoate action within this subnucleus up-regulates MC4R levels and chemogenetic stimulation of the ventrolateral VMH neurons increases locomotor activity causing weight loss in ovariectomised females (Krause et al., 2021) (Fig. 3). In contrast, the serotonin 5-HT$_{2C}$ receptors that are expressed on POMC neurons of the arcuate nucleus appear to be important functional regulators of physical activity in male but not female mice (Burke et al., 2016) (Fig. 3). Thus, the neural control of physical activity is clearly sexually dimorphic, and in females oestrogenic action to enhance physical activity appears to primarily occur within MC4R-expressing neurons of the ventrolateral VMH.

**Sexual dimorphism in adaptive thermogenesis.** In humans, the distribution and prevalence of brown adipose tissue (BAT) changes across life. BAT is most abundant in newborn infants where it is detected within the interscapular, peri-renal and neck regions (Lean et al., 1986; Lidell, 2019). During early life, BAT is essential to the maintenance of core body temperature; however, levels rapidly decline once skeletal muscle attains the ability to produce heat via shivering (Cannon & Nedergaard, 2004). Historically, dogma stipulated that unlike in rodents, in humans BAT rapidly declines during the neonatal period to reach negligible levels in adulthood (Cannon & Nedergaard, 2004). However, more recent work demonstrates that at the time of puberty there is a reinstatement of BAT, but the BAT is redistributed and is primarily located within the neck, supraclavicular and paravertebral regions (Ikeda et al., 2018; Zingaretti et al., 2009). It has been estimated that adult humans have approximately 50 g of BAT, and this can account for up to 10% of total energy expenditure (Rothwell & Stock, 1983). Moreover, positron-emission tomography–computed tomography (PET-CT) studies measuring $^{18}$F-fluoro-2-deoxy-D-glucose uptake show that BAT is most abundant in lean humans, and is reduced in individuals with obesity and T2D (Saito et al., 2009) and with ageing (Saito et al., 2009). Interestingly, a number of retrospective PET-CT studies have indicated that women have greater levels of BAT than men, which aligns with their overall greater levels of adiposity (Au-Yong et al., 2009; Cypess et al., 2009; Fletcher et al., 2020; Ouellet et al., 2011). Furthermore, recent studies have demonstrated that infra-red thermography can be used to quantify changes in BAT activity via

measurement of longitudinal changes in supraclavicular skin temperature (Fuller-Jackson et al., 2020). Studies have used thermography to demonstrate that the change in supraclavicular temperature in response to thermogenic stimuli, including cold exposure and meal intake, is greater in women than men (Fuller-Jackson et al., 2020). However, it is important to note that recent work has produced conflicting data suggesting that women have a lower supraclavicular BAT volume, but increased levels of BAT in the dorsocervical region (Fletcher et al., 2020). Therefore, further work is required to fully understand the sexually dimorphic nature of BAT in humans. Nonetheless, there is emerging evidence from rodent models that clearly identifies several mechanisms that may underpin differences in BAT activity between males and females.

Under *ad libitum* feeding conditions, female rats have a greater capacity to recruit BAT than their male counterparts (Valle et al., 2007). Earlier studies have outlined several factors that contribute to such differences including sex-related dimorphism in BAT morphology. For example, BAT mitochondria in females are dense and are composed of longer cristae than those in males (Justo et al., 2005; Rodriguez-Cuenca et al., 2002). Furthermore, the lipid profile of BAT is sexually dimorphic whereby brown adipocytes exhibit a distinct fatty acid profile, with arachidonic and stearic acids being predominant in females whereas elevated triglyceride content is evident in males (Hoene et al., 2014); this may render BAT more active in female than male rats. In addition to these morphological differences, sex steroids, in particular oestrogen, are known to alter BAT function. Indeed, the enhanced response of BAT to cold and dietary stimuli in women is correlated with circulating concentrations of $17\beta$-oestradiol (Fuller-Jackson et al., 2020).

**Role of oestrogen in the control of adaptive thermogenesis.** In ovariectomised rats, $17\beta$-oestradiol treatment causes profound weight loss compared to pair-fed controls, providing direct evidence to indicate that $17\beta$-oestradiol controls body weight via regulation of appetite and non-appetite pathways (Martinez de Morentin et al., 2014). Indeed, the aforementioned study demonstrates that $17\beta$-oestradiol treatment increases the expression of key thermogenic markers including UCP1, PPAR$\gamma$ coactivator 1$\alpha$ (PGC-1$\alpha$) and PPAR$\gamma$ in BAT (Martinez de Morentin et al., 2014), which is indicative that BAT plays a critical role in $17\beta$-oestradiol control of bodyweight.

Although earlier work indicated that oestrogens may act directly at brown adipocytes to regulate thermogenesis via increased expression of bone morphogenic protein 8b (Grefhorst et al., 2015) (Fig. 4), more recent studies in rodents provide strong evidence that oestrogens increase BAT thermogenesis via action at the brain to modulate sympathetic outflow. In mice, specific deletion of ER$\alpha$ in the VMH via either delivery of adeno-associated inhibitory viral vectors (Musatov et al., 2007) or targeted genetic deletion of ER$\alpha$ in SF1-expressing neurons (Xu et al., 2011) causes obesity. In both circumstances, weight gain is not associated with any change in food intake, but coincides with impaired BAT thermogenesis (Martinez de Morentin et al., 2014). Indeed, at least in murine models, $17\beta$-oestradiol inhibits the phosphorylation of AMP-activated protein kinase (AMPK) in neurons in the VMH, which in turn leads to activation of the sympathetic nervous system and the induction of BAT thermogenesis (Martinez de Morentin et al., 2014; Xu et al., 2011) (Fig. 4). Thus, in mice, the VMH is a central hub that is integral to relaying the effect of oestradiol on energy expenditure, where discrete subpopulations of neurons are important for relaying the effects on either physical activity (as described above) or BAT thermogenesis. It is important to note that the effects of oestrogen on adaptive thermogenesis, however, are not restricted to BAT as oestradiol also increases thermogenesis in beige adipocytes as well as skeletal muscle. In rodents, previous work has shown sexual dimorphism in the browning of white adipose tissue depots, where females exhibit greater 'browning' than males (Kim et al., 2016). In this regard, treatment with the $\beta_3$-adrenoceptor agonist CL316243 increased the expression of brown-like adipocyte markers including UCP1 in both the gonadal and inguinal adipose tissue depots of female mice, but this effect was restricted to the inguinal fat of male mice (Kim et al., 2016). Furthermore, ovarian failure and therefore oestrogen depletion induced by 4-vinylcyclohexene diepoxide treatment impaired beige cell recruitment in gonadal adipose tissue of female mice (Kim et al., 2016). Similarly, oestradiol benzoate treatment of ovariectomised ewes increases heat production in both retroperitoneal adipose tissue and skeletal muscle (Clarke et al., 2013), indicative of enhanced adipose and muscle thermogenesis, respectively. Furthermore, repeated injection of $17\beta$-oestradiol causes a sustained increase in muscle temperature in ovariectomised ewes (Clarke et al., 2013), which further supports a role for oestrogens in increasing thermogenesis across multiple tissue sites, including brown and beige adipocytes as well as skeletal muscle. Thus, oestrogen regulation of adaptive thermogenesis appears to be an important physiological component that determines sexual dimorphism in the control of energy homeostasis and metabolic function.

**Kisspeptin: a novel mediator of oestrogen induced changes in energy expenditure.** Recent work has suggested that kisspeptin is a novel regulator of body weight via modulation of energy expenditure (Tolson et al., 2014; Velasco et al., 2019). Given that kisspeptin is an integral mediator of oestrogen feedback at the level

of the hypothalamus, relaying both positive and negative feedback to the hypothalamo–pituitary–gonadal axis (Franceschini et al., 2006; Gottsch et al., 2004; Smith et al., 2005), it remains possible that the kisspeptin neurons may also be a conduit transmitting the effects of 17β-oestradiol on energy expenditure (de Roux et al., 2003; Gottsch et al., 2004). In sheep, hypothalamic Kiss1-containing neurons co-express the leptin receptor, and furthermore reciprocal connections exist between the kisspeptin and the appetite-regulating NPY and POMC neurons of the arcuate nucleus (Backholer et al., 2010). The kisspeptin receptor (KISS1R) is widely expressed within the hypothalamic brain regions, specifically within the arcuate nucleus. In female mice, genetic deletion of *Kiss1r* leads to weight gain and increased adiposity, which manifests in a reduction in energy expenditure (Tolson et al., 2014, 2016). The decrease in energy expenditure is multifactorial, where *Kiss1r* knockout mice exhibit lower locomotion, oxygen consumption and BAT thermogenesis (Tolson et al., 2016, 2020). Interestingly, this phenotype appears to be sexually dimorphic with a greater effect in females than males (Tolson et al., 2014, 2016). Furthermore, in female mice, ovariectomy control studies illustrate that weight gain

is likely to occur via both oestrogen-dependent and -independent mechanisms, where the latter is directly due to disruption in kisspeptin signalling (Tolson et al., 2014). This was further investigated in the study of Velasco et al. (2019), where *Kiss1r* expression was reinstated in the GnRH neurons of global *Kiss1r* knockout mice, which restored reproductive function and therefore sex steroid concentrations (Velasco et al., 2019). In comparison to the global *Kiss1r* knockout group, reinstatement of KISS1R in GnRH neurons obviated the increase in body weight and fat mass; however, the authors did not measure any indices of energy expenditure. Thus, further work is required to delineate the role of sex steroids in mediating changes in energy expenditure in *Kiss1r* knockout mice and the degree in which kisspeptin is involved in regulating energy expenditure, independent of sex steroid action.

## Conclusions

The control of body weight and energy homeostasis is sexually dimorphic. In rodents, females have greater satiety than males, which manifests via increased innate

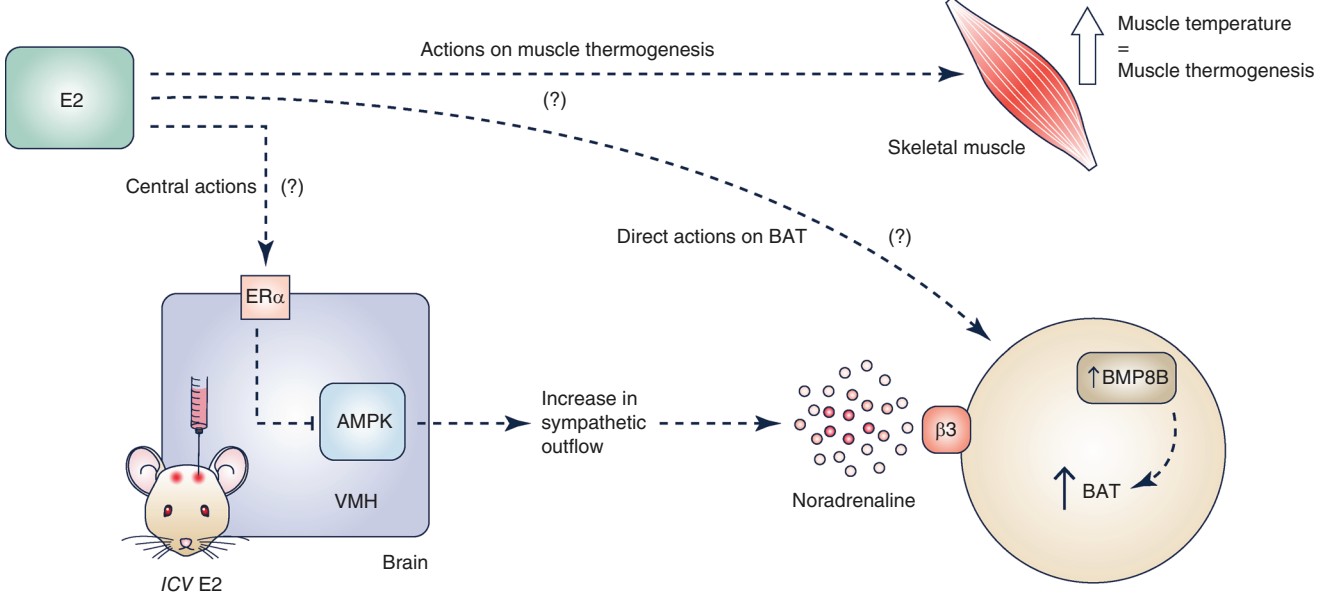

**Figure 4. The proposed mechanisms involved in thermogenic activation via 17β-oestradiol**
It is proposed that 17β-oestradiol acts via three target sites to induce thermogenesis. Mechanism 1 involves central actions within the VMH whereby 17β-oestradiol acts on ERα, which leads to the inhibition of AMPK. As a result, there is a subsequent increase in sympathetic outflow, and thus the release of noradrenaline. Mechanism 2 involves direct actions of 17β-oestradiol on BAT, which leads to the activation of endocrine factors such as BMP8B, and thus results in BAT thermogenesis. It is also speculated that 17β-oestradiol contributes to thermogenesis by acting on skeletal muscles to induce muscle thermogenesis. AMPK, AMP-activated protein kinase; BAT, brown adipose tissue; BMP8B, bone morphometric protein 8b; E2, 17β-oestradiol; ERα, oestrogen receptor α; VMH, ventromedial hypothalamus.

activity of POMC neurons. Indeed, $17\beta$-oestradiol binds to ER$\alpha$ colocalised to POMC neurons to reduce food intake. In addition to this, females have increased BAT volumes, leading to enhanced thermogenic activity. Indeed, in women, increased circulating levels of $17\beta$-oestradiol are associated with enhanced ability to activate BAT in response to thermogenic stimuli. The effect of $17\beta$-oestradiol on energy expenditure, however, is not solely attributable to changes in thermogenesis. In mice, $17\beta$-oestradiol acts at MC4R expressing neurons within the VMH to increase physical activity, and in women treatment with a GnRH agonist decreases $17\beta$-oestradiol levels causing a decline in both resting energy expenditure and physical activity. Thus, the effects of oestrogen to promote satiety and to increase overall energy expenditure are considered as key mechanisms underpinning sexual dimorphism in the control of body weight. Indeed, in animal models and in humans, females of reproductive age are less susceptible to weight gain than males, and across the menopausal transition women not only experience a change in body composition with altered adipose tissue distribution but also generalised weight gain. Thus, further understanding of the physiological mechanisms that underpin sexual dimorphism and sex steroid regulation of body weight has important ramifications for preventing weight gain in post-menopausal women. Steroidal treatments such as hormone replacement therapy, gender affirming hormonal therapy and hormonal contraceptives are widely prescribed across various stages of life. Moreover, hormonal therapies are typically prescribed across many years, which may also lead to detrimental cardiometabolic risk in itself (Diaz et al., 2012; Rossouw et al., 2002; Todd et al., 1999). Despite their wide usage, very little is known as to how these exogenous steroids impact on metabolic function and energy balance. It is therefore imperative to understand how clinically used exogenous sex steroids alter energy-balance pathways to impact on weight regulation and the altered risk of developing metabolic disease. In addition, understanding sexual dimorphism will be important to the design of new-generation metabolic and weight loss pharmacotherapies that can be tailored to males and females, and therefore improve treatment strategies for all sexes.

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

## Additional information

### Competing interests

The authors (Anne Nicole De Jesus and Belinda A. Henry) have no conflict or competing interests to declare.

### Author contributions

Anne Nicole De Jesus contributed to the preparation and editing of the manuscript. She was responsible for producing each of the Figures contained within the manuscript. Belinda A. Henry contributed to the writing and final editing of the manuscript. Both authors have read and approved the final version of this manuscript and agree to be accountable for all aspects of the work in ensuring that questions related to the accuracy or integrity of any part of the work are appropriately investigated and resolved. All persons designated as authors qualify for authorship, and all those who qualify for authorship are listed.

### Funding

This work was supported by an Advancing Women's Research Success Grant, Monash University awarded to B.A.H.

### Acknowledgements

Open access publishing facilitated by Monash University, as part of the Wiley – Monash University agreement via the Council of Australian University Librarians.

### Keywords

$17\beta$-oestradiol, energy expenditure, thermogenesis and food intake

### Supporting information

Additional supporting information can be found online in the Supporting Information section at the end of the HTML view of the article. Supporting information files available:

**Peer Review History**

