## [Peer Review History · The Journal of Physiology]

The role of estrogen in determining sexual dimorphism in energy balance

Anne Nicole De Jesus and Belinda A Henry
DOI: 10.1113/JP279501

Corresponding author(s): Belinda Henry (belinda.henry@monash.edu)

The following individual(s) involved in review of this submission have agreed to reveal their identity: Rebecca Campbell (Referee #2)

Review Timeline:

Submission Date:	02-May-2022
Editorial Decision:	14-Jun-2022
Revision Received:	18-Jul-2022
Accepted:	26-Jul-2022

Senior Editor: Laura Bennet

Reviewing Editor: Janna Morrison

Transaction Report:

Dear Dr Henry,

Re: JP-SR-2022-279501 "The role of estrogen in determining sexual dimorphism in energy balance" by Anne Nicole De Jesus and Belinda Henry

Thank you for submitting your invited Review-Symposium to The Journal of Physiology. It has been assessed by a Reviewing Editor and by 2 expert referees and I am pleased to tell you that it is considered to be acceptable for publication following satisfactory revision.

The reports are copied at the end of this email. Please address all of the points and incorporate all requested revisions, or explain in your Response to Referees why a change has not been made.

NEW POLICY: In order to improve the transparency of its peer review process The Journal of Physiology publishes online as supporting information the peer review history of all articles accepted for publication. Readers will have access to decision letters, including all Editors' comments and referee reports, for each version of the manuscript and any author responses to peer review comments. Referees can decide whether or not they wish to be named on the peer review history document.

I hope you will find the comments helpful and have no difficulty in revising your manuscript within 4 weeks.

Your revised manuscript should be submitted online using the links in Author Tasks Link Not Available. This link is to the Corresponding Author's own account, if this will cause any problems when submitting the revised version please contact us.

The image files from the previous version are retained on the system. Please ensure you replace or remove any files that have been revised. Your revised submission should include:

- A Word file of the complete text (including figure legends any Tables);
- An Abstract Figure (with legend in the Article file)
- Each figure as a separate, high quality, file;
- A full Response to Referees;
- A copy of the manuscript with the changes highlighted.
- Author profile. A short biography (no more than 100 words for one author or 150 words in total for two authors) and a portrait photograph of the two leading authors on the paper. These should be uploaded, clearly labelled, with the manuscript submission. Any standard image format for the photograph is acceptable, but the resolution should be at least 300 dpi and preferably more.

- A 'Cover Art' file for consideration as the Issue's cover image;
- Appropriate Supporting Information (Video, audio or data set https://jp.msubmit.net/cgi-bin/main.plex?form_type=display_requirements#supp).

To create your 'Response to Referees' copy all the reports, including any comments from the Reviewing Editor into a Word, or similar, file and respond to each point in colour or CAPITALS and upload this when you submit your revision.

I look forward to receiving your revised submission.

If you have any queries please reply to this email and staff will be happy to assist.

Yours sincerely,

Professor Laura Bennet
Senior Editor
The Journal of Physiology
<https://jp.msubmit.net>
<http://jp.physoc.org>
The Physiological Society
Hodgkin Huxley House
30 Farringdon Lane
London, EC1R 3AW
UK
<http://www.physoc.org>
<http://journals.physoc.org>

REQUIRED ITEMS:

-Your MS must include a complete "Additional information section" with the following 4 headings and content:

Competing Interests: A statement regarding competing interests. If there are no competing interests, a statement to this effect must be included. All authors should disclose any conflict of interest in accordance with journal policy.

Author contributions: Each author should take responsibility for a particular section of the study and have contributed to writing the paper. Acquisition of funding, administrative support or the collection of data alone does not justify authorship; these contributions to the study should be listed in the Acknowledgements. Additional information such as 'X and Y have contributed equally to this work' may be added as a footnote on the title page.

It must be stated that all authors approved the final version of the manuscript and that all persons designated as authors qualify for authorship, and all those who qualify for authorship are listed.

Funding: Authors must indicate all sources of funding, including grant numbers. If authors have not received funding, this must be stated.

It is the responsibility of authors funded by RCUK to adhere to their policy regarding funding sources and underlying research material. The policy requires funding information to be included within the acknowledgement section of a paper. Guidance on how to acknowledge funding information is provided by the Research Information Network. The policy also requires all research papers, if applicable, to include a statement on how any underlying research materials, such as data, samples or models, can be accessed. However, the policy does not require that the data must be made open. If there are considered to be good or compelling reasons to protect access to the data, for example commercial confidentiality or legitimate sensitivities around data derived from potentially identifiable human participants, these should be included in the statement.

Acknowledgements: Acknowledgements should be the minimum consistent with courtesy. The wording of acknowledgements of scientific assistance or advice must have been seen and approved by the persons concerned. This section should not include details of funding.

-Author profile(s) must be uploaded via the submission form. Authors should submit a short biography (no more than 100 words for one author or 150 words in total for two authors) and a portrait photograph of the two leading authors on the paper. These should be uploaded, clearly labelled, with the manuscript submission. Any standard image format for the photograph is acceptable, but the resolution should be at least 300 dpi and preferably more. A group photograph of all authors is also acceptable, providing the biography for the whole group does not exceed 150 words.

EDITOR COMMENTS

Reviewing Editor:

This review investigates the role of estrogen in promoting sex differences in energy balance. It is important that we understand the mechanisms that underlie sex differences.

Please assist the reviewer by numbering the lines and pages.

Introduction

When 'however' is used in the middle of a sentence it should be preceded by a semicolon and followed by a comma.

Should body conformation be morphology?

The word 'tend' is used frequently. Does this mean that there isn't a significant difference? Is there a more appropriate phrasing?

...numerous studies have shown sex differences.... Can you include some examples?

In some places the estrogens are abbreviated and in others they are not. It may improve readability if they are not abbreviated.

Section 2.0

Please indicate the species that data represent as data from many species is discussed.

Section 3.1

Please explain what the Era neurons in the hypothalamus do.

When the ewes were treated with estrogen was it daily injections or an infusion? Is a rhythm in estrogen required?

Section 4.3

BAT should be spelt out on first use.

Around the timing of puberty... or the time?

Section 3.4 appears after section 4?

Section 5.0

Should gender be sex?

..strategies for both sexes.. or for all?

Senior Editor:

Thank you for your interesting review, the reviewers have highlighted key points to address. Further, we invite you to help the reviewers and reader by considering your formatting and proof reading as per guidance given by the reviewing editor for abbreviations, and grammar etc.

REFEREE COMMENTS

Referee #1:

Summary

This is an excellent review and a pleasure to read. Sexual dimorphism is evident in both energy intake and expenditure, observed in differences in adipose tissue distribution and development of cardiometabolic diseases. The sex steroid estrogen is involved in the regulation of energy balance with implications on energy intake and expenditure. This review tackles topics of sexual dimorphism regarding the role of estrogen in energy balance through its impact on food intake and energy expenditure (resting energy expenditure, physical activity and adaptive thermogenesis).

Comments

2.0 Sexual dimorphism and role of estrogen in determining weight gain

The authors should either use more moderate language in the extrapolation of sexual dimorphism relating to developing cardiometabolic complications, considering the study referenced was performed in mice rather than a human population (consider altering to impaired inflammatory profile and impaired insulin sensitivity, which may increase risk of developing secondary cardiometabolic complications including hypertension and T2D), or make reference to some of the extensive literature on the sexual dimorphism of cardiometabolic diseases in humans between pre-menopausal women and men, and how this disparity declines as women become post-menopausal to better support their arguments.

3.0 Neuroendocrine control of food intake

Figure 1 - where is E2 produced in males?

4.2 Physical activity is regulated via estrogen in a sexually dimorphic manner

To enhance the discussion of the regulation of physical activity through estrogen, it would be worth identifying, in more explicit detail, the factors that could contribute to this increase in physical activity, e.g. increased energy?

4.0 Mechanisms of energy expenditure

The sentence regarding the mechanism of heat generation by noradrenaline activating β -adrenoceptors and then downstream UCP1 activation needs to be made clear that this is what occurs in BAT rather than WAT.

4.2 Physical activity is regulated via estrogen in a sexually dimorphic manner

The authors need to state that 5-HT_{2C} is a serotonin receptor for clarity.

4.3 Sexual dimorphism in adaptive thermogenesis

Perhaps expand on why BAT levels are greater in women, considering the authors have identified that lean body mass is greater in males and BAT is most abundant in lean individuals. Is the mass or activation of BAT greater in females (human)? This is discussed in relation to rodent studies, but not made explicitly clear for humans.

Referee #2:

This commissioned review by De Jesus and Henry discusses the role of estrogens in energy balance, and in the sexual dimorphisms of energy expenditure. The review is well written and structured and a pleasure to read. It concisely covers both early and more recent work investigating this interesting topic. I only have a few comments/ suggestions.

Specific comments:

- While the majority of the review is fairly careful about distinguishing evidence from humans vs animal models, there are sections in which this is less clear. A good proof read for this will ensure that the source of the evidence is clear for the reader.
- Are sex differences in energy balance/expenditure due solely to the activational effects of hormones like estrogen, or are there organisational effects that underpin the ways in which steroid hormones and metabolic hormones function in the adult?
- Men/male and women/female are largely discussed as binary. I wonder if there is some evidence from transgender individuals that informs this research field that could be included?
- The role of kisspeptin in E2 mediated energy balance effects is not discussed. That is fine if it is beyond the scope of the

review, but I wonder if a rationale for its absence would be worthwhile?

- In the pictorial abstract, I don't understand the crescent on top of the postmenopausal woman figure head. Is it meant to be a bun? I would remove. I also wonder if the authors might consider a more attractive colour for the post menopausal woman figure... purple maybe?
- Figure 1 is attractive, but potentially confusing. The representation of the neuropeptides as clouds (vesicles?) between the cells in the ARC and the PVN suggests volume transmission vs afferent connections and synaptic release. This is in contrast to the peripheral hormones that are indicated by direct arrows. In subsequent figures noradrenaline from the SNS is also represented as a cloud vs direct innervation to fat.
- The authors have an opportunity to suggest the most pressing/important remaining questions at the end of the review.

END OF COMMENTS

Confidential Review

02-May-2022

Response to referee and editorial comments: - JP-SR-2022-279501

Editor

Please assist the reviewer by numbering the lines and pages	We apologise for this omission and have now included both line and page numbering.
When 'however' is used in the middle of a sentence it should be preceded by a semicolon and followed by a comma.	This has been corrected throughout the text.
Should body conformation be morphology?	We have revised the word 'conformation' and used body 'composition' as this is more appropriate when describing morphology. Line 54.
The word 'tend' is used frequently. Does this mean that there isn't a significant difference? Is there a more appropriate phrasing?	We have omitted the use of 'tend' to emphasise the significant results of the cited studies.
...numerous studies have shown sex differences.... Can you include some examples?	Thank you for the suggestion. We have provided examples of sex dimorphism in the neuroendocrine control of energy balance. We have also further explained examples of mechanisms that may underpin non-steroid dependent sexual dimorphism. Lines 78-84
In some places the estrogens are abbreviated and in others they are not. It may improve readability if they are not abbreviated.	We have followed to the editor's suggestion. The full word '17 β -estradiol is now used throughout the document.
Section 2.0- Please indicate the species that data represent as data from many species is discussed.	Section 2.0 has been carefully proof read and we have included the species studied in each of the cited studies.
Section 3.1- Please explain what the ER α neurons in the hypothalamus do.	ER α -expressing neurons within the hypothalamus govern numerous physiological function, including the control of reproduction and the hypothalamo-pituitary gonadal axis, sexual behaviour, energy intake and energy expenditure. This has now been described- Lines 191-195
When the ewes were treated with estrogen was it daily injections or an infusion? Is a rhythm in estrogen required?	Estrogen treatment was obtained using an estradiol benzoate implant, which provided continuous administration. This has now been clarified. Lines 235-237

Section 4.3- BAT should be spelt out on first use.	Thank you, this has been done. Line 334
Around the timing of puberty... or the time?	This has been revised. Line 341
Section 3.4 appears after section 4?	Thank you, the typographical error has been corrected.
..strategies for both sexes.. or for all?	We have now amended this to for 'all' sexes. Line 477.

Referee 1

Section 2.0- The authors should either use more moderate language in the extrapolation of sexual dimorphism relating to developing cardiometabolic complications, considering the study referenced was performed in mice rather than a human population (consider altering to impaired inflammatory profile and impaired insulin sensitivity, which may increase risk of developing secondary cardiometabolic complications including hypertension and T2D), or make reference to some of the extensive literature on the sexual dimorphism of cardiometabolic diseases in humans between pre-menopausal women and men, and how this disparity declines as women become post-menopausal to better support their arguments.	Thank you for the suggestion. We have modified the wording to indicate that this refers to studies conducted using mice Lines Line 106-111. With regards to human studies, this has been covered in Section 1.0 and therefore section 2.0 has focussed on animal models to avoid repetition. We have also modified the subheading Line 94-95 to indicate that the work discussed has been derived from pre-clinical models.
Figure 1- Where is estrogen produced in the male?	In males, estrogen is typically produced at target tissues through the localised expression of aromatase. In light of this, we have removed the ovary from Figure 1.
Section 4.2- To enhance the discussion of the regulation of physical activity through estrogen, it would be worth identifying in more explicit detail, the factors that could contribute to this increase in physical activity, e.g. increase energy?	Thank you for the suggestion. We have expanded the discussion on Physical Activity. We hope that this addresses the Reviewers comment. Line 300-305.
Section 4.0- The sentence regarding the mechanism of heat generation by noradrenaline activating β -adrenoceptors	Thank you for the suggestion. This has been done. Line 257-258.

and then downstream UCP1 activation needs to be made clear that this is what occurs in BAT rather than WAT.	
Section 4.2 The authors need to state that 5-HT2C is a serotonin receptor for clarity	This has been done Line 326
Section 4.3 Perhaps expand on why BAT levels are greater in women, considering the authors have identified that lean body mass is greater in males and BAT is most abundant in lean individuals. Is the mass or activation of BAT greater in females (human)? This is discussed in relation to rodent studies, but not made explicitly clear for humans.	Thank you for the suggestion and we have provided further discussion on why BAT levels may be greater in women. We have highlighted that the greater BAT volume is consistent with the generalised higher adiposity in women than men. We have further highlighted that the mechanisms that may underpin this in humans is relatively unknown compared to our understanding from rodents. Line 348- 351 and Line 356-362.

Referee 2

While the majority of the review is fairly careful about distinguishing evidence from humans vs animal models, there are sections in which this is less clear. A good proof read for this will ensure that the course of evidence is clear to the reader.	Thank you for your comment. We have carefully proof read the document and provided clarification of the species studied. The changes are highlighted in yellow throughout the document.
Are sex differences in energy balance/expenditure due solely to the activational effects of hormones like estrogen, or are there organisational effects that underpin the ways in which steroid hormones and metabolic hormones function in the adult?	There may very well be organisational effects that also underpin the sex differences in energy balance. Although the review focusses on the role of estrogen we have included a small section highlighting alternative means in which sex differences may manifest. Lines78-84
Men/male and women/female are largely discussed as binary. I wonder if there is some evidence from transgender individuals that informs this research field that could be included?	There is very little information regarding the effects of gender affirming hormone therapies (GAHT) on energy balance or energy expenditure, specifically its role on adaptive thermogenesis. However, Bretherton et al. (2021) has described the effects of GAHT on adipose tissue distribution and we have now included this work Lines 58-63.
The role of kisspeptin in E2 mediated energy balance effects is not discussed. That is fine if it is beyond the scope of the review, but I wonder if a rationale for its absence would be worthwhile?	Thank you for the suggestion. We have included a small section that details the possible role of kisspeptin neurons in the regulation of energy expenditure. We highlight that this may be an alternative pathway in which estrogen can act to

	regulate body weight. Lines 420-447
In the pictorial abstract, I don't understand the crescent on top of the postmenopausal woman figure head. Is it meant to be a bun? I would remove. I also wonder if the authors might consider a more attractive colour for the post-menopausal woman figure... purple maybe?	This has been modified.
Figure 1 is attractive, but potentially confusing. The representation of the neuropeptides as clouds (vesicles?) between the cells in the ARC and the PVN suggests volume transmission vs afferent connections and synaptic release. This is in contrast to the peripheral hormones that are indicated by direct arrows. In subsequent figures noradrenaline from the SNS is also represented as a cloud vs direct innervation to fat.	Thank you for the suggestion, we have modified the Figure to demonstrate synaptic transmission.
The authors have an opportunity to suggest the most pressing/important remaining questions at the end of the review.	Thank you for the suggestion. We have provided further conclusions based on the prevalence of sex steroid-based therapies and the need to understand how these may impact on energy balance, body weight and cardiometabolic risk. Lines 467-475

Dear Dr Henry,

Re: JP-SR-2022-279501R1 "The role of estrogen in determining sexual dimorphism in energy balance" by Anne Nicole De Jesus and Belinda A Henry

I am pleased to tell you that your Symposium Review article has been accepted for publication in The Journal of Physiology, subject to any modifications to the text that may be required by the Journal Office to conform to House rules.

NEW POLICY: In order to improve the transparency of its peer review process The Journal of Physiology publishes online as supporting information the peer review history of all articles accepted for publication. Readers will have access to decision letters, including all Editors' comments and referee reports, for each version of the manuscript and any author responses to peer review comments. Referees can decide whether or not they wish to be named on the peer review history document.

The last Word version of the paper submitted will be used by the Production Editors to prepare your proof. When this is ready you will receive an email containing a link to Wiley's Online Proofing System. The proof should be checked and corrected as quickly as possible.

All queries at proof stage should be sent to tjp@wiley.com

The accepted version of the manuscript is the version that will be published online until the copy edited and typeset version is available. Authors should note that it is too late at this point to offer corrections prior to proofing. Major corrections at proof stage, such as changes to figures, will be referred to the Reviewing Editor for approval before they can be incorporated. Only minor changes, such as to style and consistency, should be made a proof stage. Changes that need to be made after proof stage will usually require a formal correction notice.

Are you on Twitter? Once your paper is online, why not share your achievement with your followers. Please tag The Journal (@jphysiol) in any tweets and we will share your accepted paper with our 22,000+ followers!

Yours sincerely,

Professor Laura Bennet
Senior Editor
The Journal of Physiology
<https://jp.msubmit.net>
<http://jp.physoc.org>
The Physiological Society
Hodgkin Huxley House
30 Farringdon Lane
London, EC1R 3AW
UK
<http://www.physoc.org>
<http://journals.physoc.org>

Editor Comments:

Thank you for addressing the reviewers' comments and revising the paper.

REFEREE COMMENTS:

Referee #1:

No further comments.

Referee #2:

The authors have done a thorough job of responding to my comments. The figures are much improved. I have no further concerns. Congratulations on a very nice review.

*** IMPORTANT NOTICE ABOUT OPEN ACCESS ***

To assist authors whose funding agencies mandate public access to published research findings sooner than 12 months after publication The Journal of Physiology allows authors to pay an open access (OA) fee to have their papers made freely available immediately on publication.

You will receive an email from Wiley with details on how to register or log-in to Wiley Authors Services where you will be able to place an OnlineOpen order.

You can check if your funder or institution has a Wiley Open Access Account here <https://authorservices.wiley.com/author-resources/Journal-Authors/licensing-and-open-access/open-access/author-compliance-tool.html>

Your article will be made Open Access upon publication, or as soon as payment is received.

If you wish to put your paper on an OA website such as PMC or UKPMC or your institutional repository within 12 months of publication you must pay the open access fee, which covers the cost of publication.

OnlineOpen articles are deposited in PubMed Central (PMC) and PMC mirror sites. Authors of OnlineOpen articles are permitted to post the final, published PDF of their article on a website, institutional repository, or other free public server, immediately on publication.

Note to NIH-funded authors: The Journal of Physiology is published on PMC 12 months after publication, NIH-funded authors DO NOT NEED to pay to publish and DO NOT NEED to post their accepted papers on PMC.

1st Confidential Review

18-Jul-2022